# NorMASS: A normative MAS-based modeling approach for simulating incentive mechanisms of Q&A communities

Yi Yang[1,2,3]*, Xinjun Mao[1,3], Shuo Yang[4], Menghan Wu[1,3]

**1** College of Computer, National University of Defense Technology, Changsha, Hunan, China, **2** College of Information Science and Engineering, Hunan Women's University, Changsha, Hunan, China, **3** Key Laboratory of Complex Systems Software Engineering, National University of Defense Technology, Changsha, Hunan, China, **4** College of Systems Engineering, National University of Defense Technology, Changsha, Hunan, China

* yangyi@nudt.edu.cn

**Data Availability Statement:** The dataset used in this research is openly accessible via https://www.github.com/snryou/NorMASS.

**Funding:** XM received funding for this research from the National Key Research and Development Project of China under Grant 2018YFB1004202.

## Abstract

Incentive mechanisms steer users in Q&A communities to achieve community goals, which need to be cautiously reviewed and revised before actual industrial application. Simulating incentive mechanisms is significant for predicting how changes in incentive mechanisms will affect community emergence, such as user answering patterns. However, due to the complexity of Q&A communities, the challenge faced by simulating incentive mechanisms lies in the difficulty of establishing micro-macro connections in the communities to simulate their emergence. To fill this gap, this paper proposes a Normative Multi-Agent System based Simulation (NorMASS) approach to simulate community emergence. The NorMASS models a Q&A community as a normative multi-agent system and adopts agents to formally express community users. Moreover, the approach provides an open-source simulator with a data generator to simulate community emergence. An evaluation of the NorMASS comparing simulation emergence with the counterpart of an actual community demonstrates that the proposed approach provides an effective solution for simulating incentive mechanisms of Q&A communities, with a similarity of 80% or above.

## 1. Introduction

Question and answer (Q&A) communities such as Stack Overflow.com (SO) that rely heavily on their users to contribute massive amounts of content lead to great success. In these communities, their managers need to formulate and implement incentive mechanisms such as reputation, badge, and privilege to meet a range of objectives, including encouraging users to participate more. These incentive mechanisms need to be cautiously reviewed and revised to align them with community goals before actual industrial application.

Simulating the incentive mechanism is significant for predicting the impact of an incentive mechanism reform and adjustment and making sure that such a change is a boost to the community. In Q&A communities, this type of simulation generally belongs to equation-based

The funders had no role in study design, data collection and analysis, decision to publish, or preparation of the manuscript.

**Competing interests:** The authors have declared that no competing interests exist.

modeling (EBM) [1]. EBM applies a set of statistical or mathematical equations at the macro level to articulate the relationships between communities' incentive mechanisms and their macro phenomena (e.g., user answering patterns) [2, 3]. The simulation results are indicative of how the incentive mechanisms would affect actual communities at the macro level. For example, Gao et al. [4] use a game-theoretic model that simulates and predicts Q&A community users' answering-voting pattern under the presence of a reputation mechanism. Goes et al. [5] draw on goal-setting and status hierarchy theories to examine the effect of a status-based incentive hierarchy on user contributions in online knowledge exchange. Jing et al. [6] construct a fixed-effect specification-regression model to explore the moderation effects of monetary incentives on physician behavior. Papoutsoglou et al. [7] used multivariate cluster analysis to model and simulate the effect of the badges gamification mechanism on personality traits.

However, existing EBM approaches, e.g., the game-theoretic model [4], the regression model [6], and the hidden Markov model [8], are difficult to establish micro-macro connections in the communities to simulate their macro phenomena because Q&A communities are typical complex social-technical systems [9]. Q&A communities are a kind of virtual space where massive autonomous users scattered in different regions conduct questioning and answering with the support of network information technology [10]. The users are heterogeneous, owning different reputations, preferences, behavior patterns. There exist massive nonlinear interactions and relationships in these communities. For example, users may make multiple decisions, including answering and voting, on peers' questions. These massive interactions can result in special macro phenomena that cannot be simply derived from the summation of individuals' behaviors, which we call emergence [11]. Besides, in EBM approaches, the characteristics of a population are generally averaged [12]. Thus, the approaches using a homogeneous population undergo problems describing the nonlinear interactions among massive heterogeneous community users and the resulting emergence [13–15].

Considering the challenges above, we develop a bottom-up modeling approach named NorMASS (Normative Multi-Agent System [16] based Simulation) for simulating the incentive mechanism of Q&A communities in this paper, focusing on reputation mechanisms. These mechanisms are among the most used incentive mechanisms adopted by Q&A communities, providing users with virtual points that display skills and achievements in these communities [5]. Reputation mechanisms distinguish the set of users as "status classes" that grant them some privileges to access more content based on particular metrics [17, 18]. In the rest of this paper, we, therefore, take incentive mechanism to mean "reputation mechanism".

We adopt a normative Multi-Agent System (MAS) based modeling approach in this paper for several reasons. First, the normative MAS approach can naturally represent heterogeneous autonomous individuals in communities. Hence, it can simulate nonlinear interactions among individuals at the micro level and the resulting emergence at the macro level [19, 20]. Next, the approach allows fine-tuning of parameters to explore the impact of specific incentive mechanisms on community emergence [21]. Finally, as normative MAS based approaches being generative, we can easily establish the causality for specific incentive mechanisms and community emergence [13, 22].

In summary, this work makes the following contributions:

1. **Normative MAS-based formal simulation model.** We use a normative MAS to represent Q&A communities. It explicitly represents active agents (community users), passive objects (e.g., questions and answers), and norms (incentive mechanisms) in the system. This formalism articulates the role of the NorMASS in simulating the micro Q&A process, incentive regulation, and macro emergence.

2. **Formal representation of agents in NorMASS**. We design a set of formal formulas to express agent characteristics according to users' statistics of the real world. This enables us to specify the characteristics of heterogeneous agents' attributes and behaviors in our normative MAS-based approach.

3. **Design and derivation of simulation data generator**. We propose a data simulation algorithm and develop a simulation data generator based on community users' statistical information to overcome the lack of complete and real community data, or to save the time and effort for data gathering.

4. **Development of the open-source incentive mechanism simulator**. We develop an open-source incentive mechanism simulator that enables us to compare simulation emergence with the counterpart of the real community to evaluate the NorMASS (https://www.github.com/snryou/NorMASS).

The remainder of the article is organized as follows. Section 2 introduces the related work. Section 3 describes an exemplar problem of the Stack Overflow community. Sections 4 and 5 describe our NorMASS and its evaluation, respectively. Section 6 concludes the paper.

## 2. Related work

### 2.1. Emergence simulation

Our work is closely related to the use of the Agent-Based Modeling (ABM) approach for studying the emergence of virtual communities. Gatti et al. [23] developed an agent-based model to examine message diffuse patterns of users in Micro-blogging communities. Yu et al. [24] adopted the agent simulation method to explore the impact of the strategy for answering diverse selection questions on website performance by users of varied types. Erik et al. [25] adopted an ABM method to simulate the knowledge cooperation behavior of users in the Q&A community of SAP (https://answers.sap.com). Jiang et al. [26] used an ABM method to simulate the impact of click position on user answer behavior.

These studies discuss user attributes (e.g., activity and preference) and interaction rules, providing suggestions for our model design. Our work differs from the literature above in that our simulation of emergence takes into account the factor of incentive mechanisms.

### 2.2. Modeling social norms

Our work is also related to the literature on modeling social norms. Aldewereld et al. [27] presented the concept of group norms to regulate three sets of agents. Viana et al. [28] created a model language to support the systematic design of adaptive normative multi-agent systems. Brito et al. [29] situated norms in a hybrid, interactive, normative multi-agent system to provide a context-aware crisis regulation. Bulling et al. [30] proposed a concrete executable specification language to study and analyze the effect of norms and sanctions on the behavior of rational agents. Dell'Anna et al. [31] proposed a runtime mechanism for the automated revision of norms to control and coordinate the behavior of individual agents in multi-agent systems.

Our work differs from the literature above in that our work focuses on modeling social norms (e.g., incentive mechanisms) from the real world, not a conceptual multi-agent system.

## 3. Exemplar problem description of SO

The Stack Overflow community (SO community) can be represented in Fig 1. The community we study has tens of millions of users and posts (questions and answers), and hundreds of

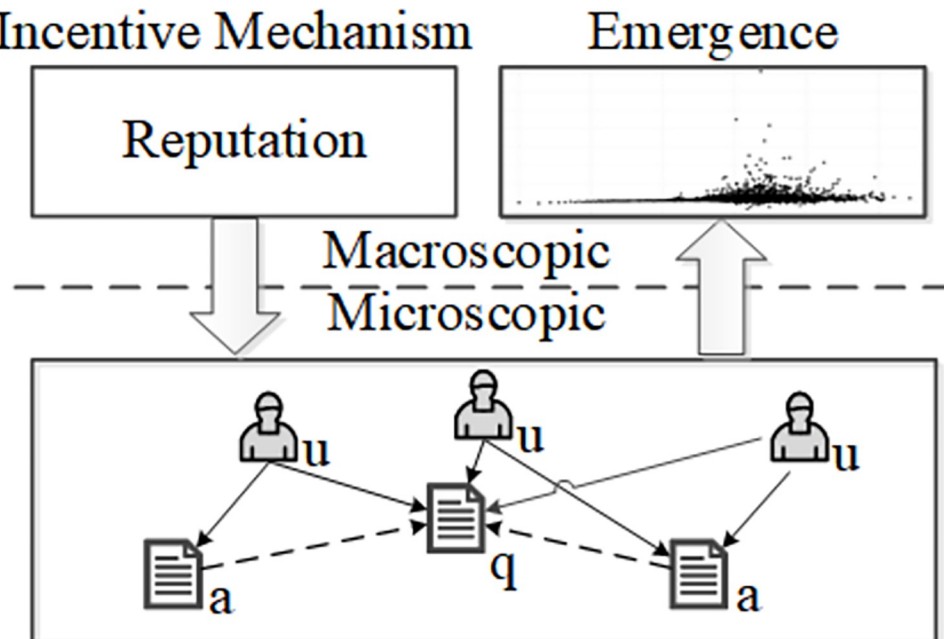

**Fig 1. Schematic of the SO community.** The symbols $u$, $q$, and $a$ represent the community user, the question, and the answer, respectively. Edges between users and posts represent interactions (questioning, answering, and voting), while edges between posts represent the correlations between a question and its answers. The interactions between users are nonlinear under the regulation of SO incentive mechanisms. These massive nonlinear interactions lead to the SO macro emergence.

millions of user-post interactions over a 32-month window from Jan. 2017 to Sep. 2019. User interactions are indirectly achieved through their posts. For example, a user may create a question, and others create answers to the question. In addition, the quality of the question and its answers is evaluated by peers' voting (upvoting and downvoting) in SO.

The interaction among users can influence their reputation points according to the SO reputation mechanism (https://stackoverflow.com/help/whats-reputation), as shown in Table 1. For example, the community rewards a user with ten points when his/her question receives an upvote. In contrast, it may be punished by deducting two points for a downvoted question. Meanwhile, a voter may be inflicted a penalty of one point in order to ensure it votes down a low-quality question only. SO users are heterogeneous and have different attributes (e.g. reputation, preference), thus the interactions among them are nonlinear. For example, users may make several decisions to questions of different answer numbers for obtaining more reputation. These massive non-linear interactions can lead to SO macro emergence.

It is necessary to predict community emergence resulting from incentive mechanism reform to avoid adverse side effects. For example, Stack Overflow reformed its reputation mechanism by uplifting the reward points for upvoted questions on Nov. 13, 2019, and the

**Table 1. Reputation update rules in SO.**

| Rule | Action | Reputation change |
|---|---|---|
| 1 | Question is upvoted | +10 to owner |
| 2 | Question is downvoted | -2 to owner |
| 3 | Answer is upvoted | +10 to owner |
| 4 | Answer is downvoted | -2 to owner |
| 5 | Downvote an answer | -1 to voter |

mechanism reform has been widely debated (https://meta.stackoverflow.com/questions/391250/upvotes-on-questions-will-now-be-worth-the-same-as-upvotes-on-answers). Some users did not agree with the incentive mechanism reform and declared reducing their answering or even leaving the community. Thus, communities need to minimize the negative effects due to mechanism reforms. To facilitate decision-making on reforming incentive mechanisms, it is necessary to build a decision support model. One such risk assessment model can promote community managers in daily operations by inferring the risk of each incentive mechanism and deciding which incentive mechanism to use.

# 4. NorMASS

## 4.1. Overview

Fig 2 illustrates the proposed approach NorMASS. In Step 1, Modeling incentive mechanisms, we describe the incentive mechanisms by interpreting the web pages of SO incentive mechanisms (https://stackoverflow.com/help/whats-reputation). The step produces two outputs. One is a QA model of the incentive mechanism context abstracted as a MAS-based model. Another is a mechanism model expressing the realization of the incentive mechanism employing the form of first-order logic.

In Step 2, we enrich the QA model using the statistical data of community users, on which the mechanism simulation needs to be executed. We collect this information by applying statistics to the actual community data set. Our proposed formula provides a theoretical basis for the establishment of the rationality of the model for capturing the statistical characteristics of community users.

In Step 3, based on this statistical information, we design a reasonable algorithm for data collecting and generate the required simulation data in the following steps. The data is

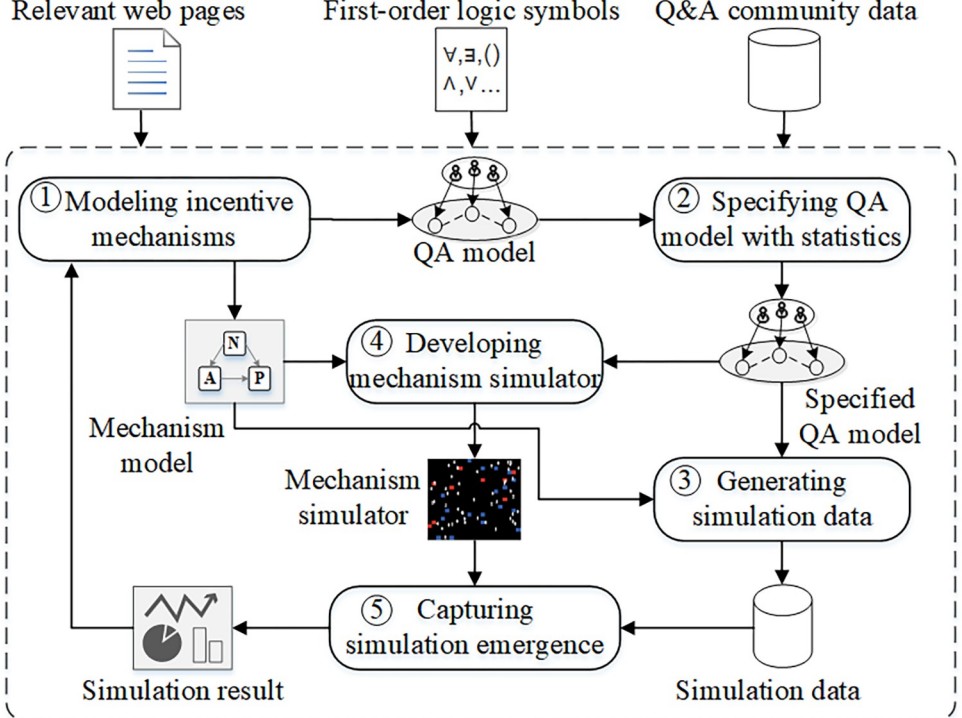

**Fig 2. Simulation approach overview.**

employed by the simulator in Step 5 to simulate the state of the real community on a specific date.

In Step 4, Developing mechanism simulator, we develop an incentive mechanism simulation system based on the proposed model for running the simulation in Step 5.

In Step 5, we run the incentive mechanism simulator on the generated data to capture macro emergence that is used to evaluate the performance of our approach.

We produce an open-source reusable tool that can present simulation results to the user to check against the impact of incentive mechanisms on the real community. If the results are inconsistent, the incentive mechanism model can be modified and the simulation process repeated. Furthermore, with the tool, community managers and practitioners can compare the simulation results of multiple incentive mechanisms on the same simulation data to choose the incentive mechanism best aligning with the real community's goals.

## 4.2. Modeling incentive mechanisms

To execute incentive mechanism analysis, incentive mechanisms need to be precisely interpreted and expressed. For this reason, we develop a normative MAS-based modeling approach for specifying community incentive mechanisms. The approach yields two main outputs, as presented in Fig 2: a QA model and a mechanism model. The representations are written in the first-order logic.

We consider the reputation mechanism of SO as an example. As shown in Table 1, when a user's posts are voted up by its peers, certain reputation is gained on the part of it. Otherwise, certain reputation is lost on its part. Hence, we can conclude that a Q&A community consists of users, their posts, and a set of incentive rules. Based on the idea of normative MAS, we model a Q&A community as a tuple $NMAS=<QA, NORM>$, in which, $NMAS$ represents a Q&A community, $NORM$ and $QA$ represent the incentive mechanism and its context in the community, respectively.

**4.2.1. QA model.** The objects regulated by an incentive mechanism are community users and their environment. Thus we model the context of incentive mechanisms in Q&A communities as $QA=<AG, POST>$. Here, $AG$ is the set of autonomous agents that represent Q&A community users. Each agent ($ag{\in}AG$) has its own unique attributes and some behaviors including creating posts. The environment of agents refers to the context agents interact. The interactions among community users are based on their posts. Thus we define the environment of agents as $POST$ which represents the set of posts of agents.

*Agent attributes.* We focus on users' attributes related to incentive mechanisms, including reputation, posting and voting activeness, questioning preference, and voting preference. In addition, users follow incentive rules to participate in the community. Accordingly, we define an agent $ag$ as a tuple $ag=<rep, pac, vac, qr, ur, RULE>$ where

- **rep** is an agent's reputation. Just like in a real community, an individual's reputation is not less than one, i.e. $rep{\geq}1$.

- **pac** is the average number of agent $ag$'s daily posts, which represents the agent's posting activeness. Here, $pac{\geq}0$.

- **vac** is the average number of agent $ag$'s daily votes, which represents the agent's voting activeness. Here, $vac{\geq}0$.

- **qr** is an agent's question rate that indicates the agent's questioning preference. It is measured by the rate of the agent's questions to its total posts (questions and answers). Here, $qr{\in}[0, 1]$. Accordingly, the agent's answer rate is $1$-$qr$.

- **ur** is an agent's upvote rate that indicates the agent's upvoting preference. It is measured by the rate of the agent's upvotes to its total votes (upvotes and downvotes). Here, $ur \in [0, 1]$. Accordingly, the agent's downvoting rate is $1\text{-}ur$.

- **RULE** represents the behavior rules followed by agents as introduced in Section 4.2.3.

**4.2.2. Environment of agents *POST*.** The environment of agents *POST* is the union of agents' question set and answer set, i.e. *POST=Q∪A*, where *Q* and *A* represent agents' question set and answer set, respectively. For each post *p∈POST*, we define *p=<type, ag>* where

- **type** is the type of post *p*. Here, *type∈* [0, 1]. 0 and 1 represent a question and an answer, respectively.

- **ag** is the agent creating post *p*.

In addition, there are some relations between questions and answers in *POST*. Here, we use *QRA={<q, a>|q∈Q∧a∈A}* to represent the relations between questions *Q* and answers *A*. *<q, a>* represents an answer *a* in *A* is the answer of a question *q* in *Q*.

**4.2.3. Agent interaction rule set *RULE*.** In this paper, we focus on five types of agent behavior related to their reputation: answering, upvoting, downvoting, and updating. Accordingly, we define five rules to constrain agents' behavior. To explain our defined rules and semantics in *NMAS*, we make use of some symbols that are shown in Table 2. In addition, we use *R* and *Z* to indicate the sets of reals and integers, respectively.

- **Questioning rule.** Q&A community users ask questions for seeking information on solving their technology problems. Here, we use *pq* to represent the extent of agent *ag*'s current demand for information. *ag*'s question rate *qr* denotes the overall degree of *ag*'s demand for information, which can be treated as a threshold. As shown in Eq 1, when *ag*'s *pq* is greater than its question rate *qr*, the agent asks a question.

$$\forall ag \in AG \exists pq, qr \in [0,1] \exists q \in Q$$

$$(PQ(ag, pq) \wedge QR(ag, qr) \wedge (pq \geq qr) \rightarrow CR(ag, q)). \tag{1}$$

**Table 2. Predicts of the proposed model.**

| Predicts | Definitions |
|---|---|
| CR(*ag*, *p*) | Post *p* is created by agent *ag*. |
| DV(*ag*, *p*) | Agent *ag* downvotes post *p*. |
| PA(*ag*, *pa*) | Agent *ag*'s answering probability is *pa*. |
| PQ(*ag*, *pa*) | Agent *ag*'s questioning probability is *pq*. |
| PD(*ag*, *pd*) | Agent *ag*'s downvoting probability is *pd*. |
| PU(*ag*, *pu*) | Agent *ag*'s upvoting probability is *pu*. |
| PAC(*ag*, *pac*) | Average number of agent *ag*'s daily posts is *pac*. |
| QR(*ag*, *qr*) | The question rate of agent *ag* is *qr*. |
| REP(*ag*, *rep*) | The reputation points of agent *ag* is *rep*. |
| RE(*ag*, *pt*) | Agent *ag* earns *pt* points. |
| UV(*ag*, *p*) | Agent *ag* upvotes post *p*. |
| VAC(*ag*, *vac*) | Average number of agent *ag*'s daily votes is *vac*. |
| UR(*ag*, *ur*) | The upvote rate of agent *ag* is *ur*. |

- **Answering rule**. Similar to the questioning rule, when agent *ag*'s current answering probability *pa* is greater than its answer rate 1-*qr*, agent *ag* answers a question. The answering pattern of agents is described as Eq 2.

$$\forall ag \in AG \exists pa, qr \in [0,1] \exists q \in Q \exists a \in A$$

$$(PA(ag, pa) \land QR(ag, qr) \land (pa \geq 1 - qr) \rightarrow CR(ag, a) \land (< q, a > \in QRA)). \tag{2}$$

- **Upvoting rule and downvoting rule.** We assume that if an agent's current upvoting probability *pu* is equal to or greater than its upvote rate *ur*, it upvotes a post. In contrast, if an agent's current downvoting probability *pd* is equal or greater than 1-*ur*, it downvotes the post. The voting rules are described as Eqs 3 and 4.

$$\forall ag \in AG \exists pu, ur \in [0,1] \exists p \in POST$$

$$(PU(ag, pu) \land UR(ag, ur) \land (pu \geq ur) \rightarrow UV(ag, p)). \tag{3}$$

$$\forall ag \in AG \exists pd, ur \in [0,1] \exists p \in POST$$

$$(PD(ag, pd) \land UR(ag, ur) \land (pd \geq 1 - ur) \rightarrow DV(ag, p)). \tag{4}$$

- **Updating rule.** Steered by incentive mechanisms, community users' internal or external motivation strength changes with their reputations. For example, higher reputation users are more willing to vote on others' posts than lower peers [32]. Hence, we assume that when an agent earns or loses some points *pt*, it automatically updates its attributes, as shown in Eq 5. Here, *pt* is an integer. $f_2, f_3, f_4$, and $f_5$ are some formulas for evaluating the effects of users' reputations on their other attributes.

$$\forall ag \in AG \exists rep \geq 1 \exists pac, vac \geq 0 \exists qr, ur \in [0,1] \exists pt \in Z$$

$$\left( \left( \begin{array}{c} REP(ag, rep) \land PAC(ag, pac) \land \\ VAC(ag, vac) \land QR(ag, qr) \land \\ UR(ag, ur) \land RE(ag, pt) \end{array} \right) \rightarrow \left( \begin{array}{c} REP(ag, rep + pt) \land PAC(ag, f_2(rep + pt)) \land \\ VAC(ag, f_3(rep + pt)) \land QR(ag, f_4(rep + pt)) \land \\ UR(ag, f_5(rep + pt)) \end{array} \right) \right). \tag{5}$$

**4.2.4. Mechanism model.** From the insights of the community incentive mechanism described in Table 1, we define two norms in *NORM*: reward norm and penalty norm.

- **Reward norm.** As shown in Eq 6, when agent $ag_1$ votes up post *p* of agent $ag_2$, they are rewarded $pt_1$ and $pt_2$ points, respectively.

$$\forall ag_1, ag_2 \in AG \exists p \in POST \forall pt_1, pt_2 \in Z$$

$$(UV(ag_1, p) \land CR(ag_2, p) \rightarrow RE(ag_1, pt_1) \land RE(ag_2, pt_2)). \tag{6}$$

- **Penalty norm.** On the contrary, when agent $ag_1$ votes down post $p$ of agent $ag_2$, they are penalized $pt_3$ and $pt_4$ points, respectively (see Eq 7).

$$\forall ag_1, ag_2 \in AG \exists p \in POST \forall pt_3, pt_4 \in Z$$

$$(DV(ag_1, p) \wedge CR(ag_2, p) \rightarrow RE(ag_1, pt_3) \wedge RE(ag_2, pt_4)). \tag{7}$$

## 4.3. Specifying QA model with statistics

In this section, we present some equations to enrich the QA model from three aspects: agent reputation distribution, the relations between agents' reputation and the other attributes, and agents' behavior probability. The statistics for specifying the QA model come from our observation of Q&A communities and some other researchers' reports.

Our analysis of Q&A community users' characteristics is based on data collected from the language communities of Stack Overflow whose users are aggregated. The questions of Q&A communities are generally tagged with program languages such as Java and Python [33]. These tags partition a Q&A community into different language communities where intra-group interactions are more frequent than inter-group interactions [34], reflecting the aggregation of users. This structure of the Q&A community enables us to explore more conveniently and accurately the characteristics of the whole Q&A community with its certain language community [35]. Hence, we specific agents' attributes and behaviors characteristics by analyzing users' attributes and behavior of the Java language community of SO where there is a rich offering of user data. The data are from the SO community between December 2008 and September 2019.

**4.3.1. Expressing agent reputation distribution.** The success of Q&A communities depends mainly on the contribution of a small number of expert users owing to a high reputation, while most of users have a low reputation due to fewer contributions [36, 37]. As shown in Fig 3, the distribution of SO users with different reputations obeys a power-law distribution based on our observation from SO data. Thus, we use Eq 8 to describe agent reputation distribution. Here, $f_1(rep)$ is the number of agents with *rep* points. The parameters $a_1$ and $b_1$ are two

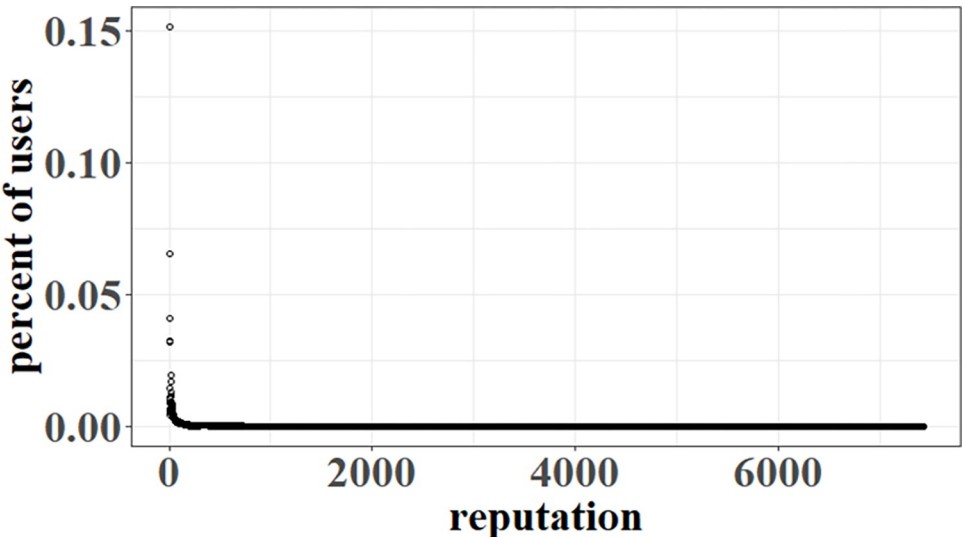

**Fig 3. SO users' reputation distribution.**

reals that fit the distribution, i.e., $a_1$, $b_1 \in R$.

$$f_1(rep) = a_1 * rep^{b_1}. \tag{8}$$

### 4.3.2. Expressing agent attribute relationship. *Contribution activeness.* Movshovitz-Attias et al. [38] found that users of higher reputations create more posts than users of lower reputations on average. From the upper left of Fig 4, we can see that the SO users' daily posts increase with their reputation points. The same applies to users' voting, as shown at the bottom left of Fig 4. Thus, we use Eqs 9 and 10 to describe the relations between agents' reputation and activeness in contribution. Here, $a_2$, $b_2$, $a_3$, $b_3 \in R$.

$$f_2(rep) = a_2 * rep + b_2. \tag{9}$$

$$f_3(rep) = a_3 * rep + b_3. \tag{10}$$

*Contribution preference.* From the right part of Fig 4, we can see SO users' question rate and upvote rate decrease with their points. Therefore, we use Eqs 11 and 12 to describe the relations between agents' reputation points and contribution preference. Here, $a_4$, $b_4$, $a_5$, $b_5 \in R$.

$$f_4(rep) = a_4 * rep + b_4. \tag{11}$$

$$f_5(rep) = a_5 * rep + b_5. \tag{12}$$

We acknowledge that linear equations do not fit the relationships between users' attributes very well. However, they can reflect the general trends of the correlations between users'

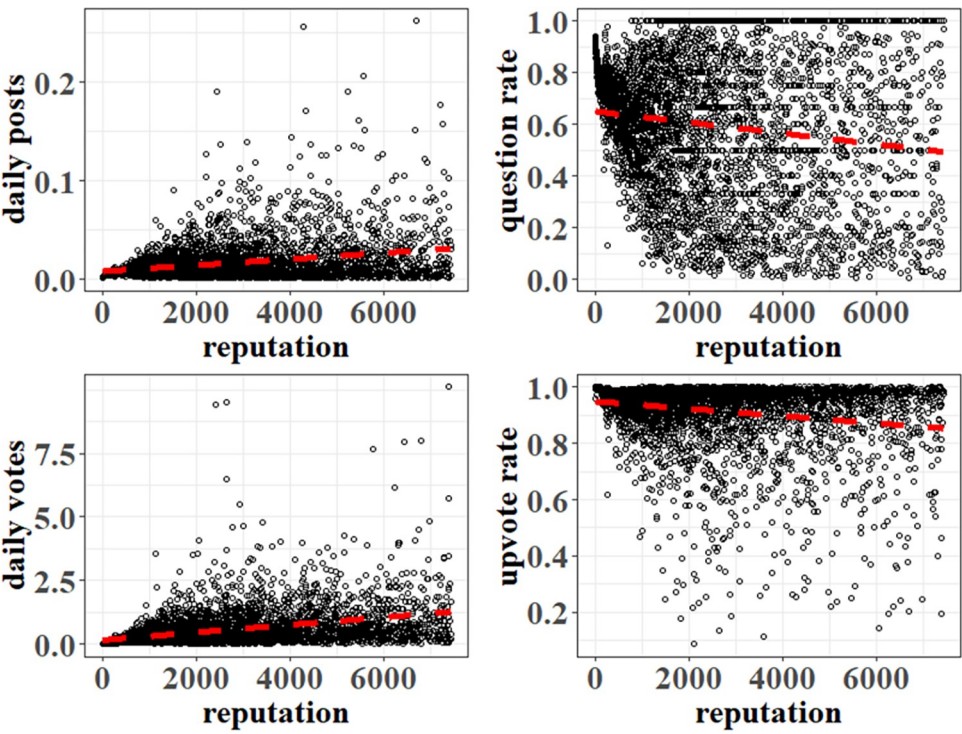

**Fig 4. The relationship between SO users' attributes.**

attributes, sufficient to meet our needs for simulating the effects of a reputation mechanism on its community.

**4.3.3. Expressing agent behavior probability.** Users' behavior probability represents the extent of their current needs. Information need is found to be the most important motivation for users to ask questions [39]. While users' voting is considered a response due to reciprocity [40]. Both behaviors are stochastic and least affected by incentive mechanisms. Thus, we employ some random numbers between 0 and 1 for expressing two users' behavior probabilities at a certain time, i.e. $pq, pu \in [0, 1]$.

Differently, agents' current answering probability is determined by the attributes of the question they intend to answer. Based on Yang et al.'s work [32], users are significantly regulated by the reputation mechanism to answer questions. The more answers a question has, the more difficult a user receives upvotes and points on the part of a user, it is less probable that the user answers. Apart from that, according to [41], the more delayed a user's answer to a question, the less probable it is that it receives upvotes and points, and it is less probable that the user answers. Here, we use the function $pan(n)$ to describe an agent's current probability of answering a question having $n$ answers, and the function $pat(t)$ to represent an agent's current probability of answering a question asked $t$ days ago. The time length of a question is also called question-age. Thus, an agent's current probability of answering $pa$ can be measured by $pan(n)^*pat(t)$.

## 4.4. Generating simulation data

In this section, we design a reasonable algorithm to generate the required simulation data for two reasons. One is that access to real data is partly restricted in Q&A communities. For example, it is difficult to obtain complete voting information from SO. In addition, when a new incentive mechanism is being introduced, there may be no real data for simulation. Another is that the simulated data enables fast and efficient experiments, saving the time and effort for data gathering. An overview of this process is shown in Fig 5. The inputs for the process are a QA model specified with the equations of Section 4.3 and the incentive mechanism to simulate. The parameters related to Eqs 1–5 in the process are determined by the statistics of the simulated community. First, we use Eq 1 to generate the distribution of agents of different reputations. Here, the total number of agents is determined by the user number of the real community. Second, we modify some other attributes such as post-activity $pac$ based on Eqs 2–5. Finally, we generate a mechanism instance and use it to check agent attributes. For example, an agent's reputation is not less than one.

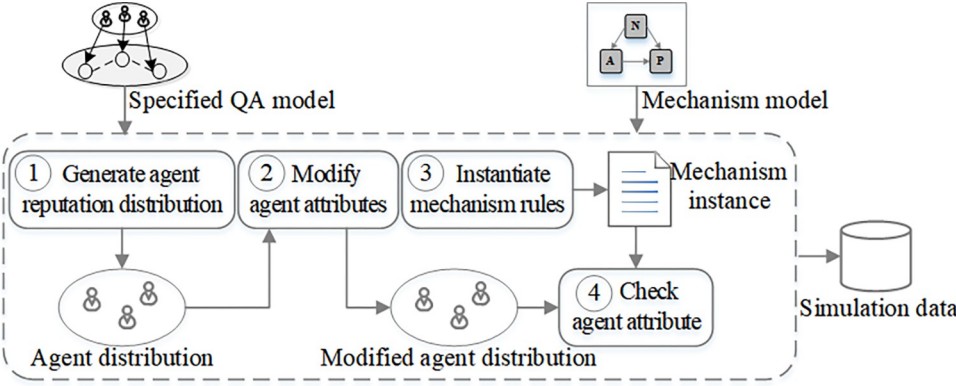

**Fig 5. Overview of simulation data generation.**

---

**Algorithm 1 : Simulation Data Generation**

**Input:** (1) $QA$; (2) $PT$; (3) $agNum$
**Output:** $AG$, $NORM$

1: $AG \leftarrow \emptyset$
2: $rep \leftarrow 0$
3: $aNum \leftarrow agNum$
4: **while** $aNum > 0$ **do**
5: $rep \leftarrow rep + 1$
6: $repNum \leftarrow \mathrm{cal}(rep, agNum, QA)$
7: **if** $repNum > 1$ **then**
8: $repAg \leftarrow \mathrm{create}(repNum, rep)$
9: $aNum \leftarrow aNum - repNum$
10: **else**
11: $repAg \leftarrow \mathrm{create}(1, \mathrm{random}(MAXPOINT))$
12: $aNum \leftarrow aNum - 1$
13: **end if**
14: $AG \leftarrow AG \cup repAg$
15: **end while**
16: $AG \leftarrow \mathrm{modify}(AG, QA)$
17: $NORM \leftarrow \mathrm{instance}(PT)$
18: $AG \leftarrow \mathrm{check}(AG, NORM)$
19: **return** $AG$, $NORM$

**Fig 6. Simulation data generation.**

The simulation data generation algorithm is shown in Fig 6. The algorithm takes the following inputs: (1) the specified QA model $QA$. (2) Reputation mechanism parameter set $PT$. (3) the number of agents $agNum$. The algorithm has four main parts as explained below.

1. **Generate agent reputation distribution** (L. 1-15). Lines 1-3 initialize the related parameters. Lines 5-9 calculate agent numbers with different reputations (less than 200) and create them. Lines 10-11 crate randomly few agents of high reputation that represent the expert users of Q&A communities.

2. **Modify agent attributes** (L. 16). After creating the agents, we modify their attributes using Eqs 2–5 in the specified QA model.

3. **Instantiate mechanism rule** (L. 17). The algorithm instantiates the mechanism model by using the input point set $PT$.

4. **Check agent attributes** (L. 18). After modifying the agents' attributes, we check whether their attributes are reasonable. For example, agents' reputations are not less than 1 and their $qr$, $ur$ are not greater than 1.

## 4.5. Developing mechanism simulator

Our mechanism simulator is based on the multi-agent modeling framework NetLogo [42]. Having been implemented, the simulator is shown in Fig 7. Little white figures are turtles that represent autonomous agents of NorMASS. The turtles are a type of active entity capable of asking questions, answering and voting, as well as updating their own states. The blue blocks represent questions and the red blocks represent answers. These blocks are a kind of passive entity (patches) representing the environment of the turtles, which can only be created and modified by turtles without their own behavior.

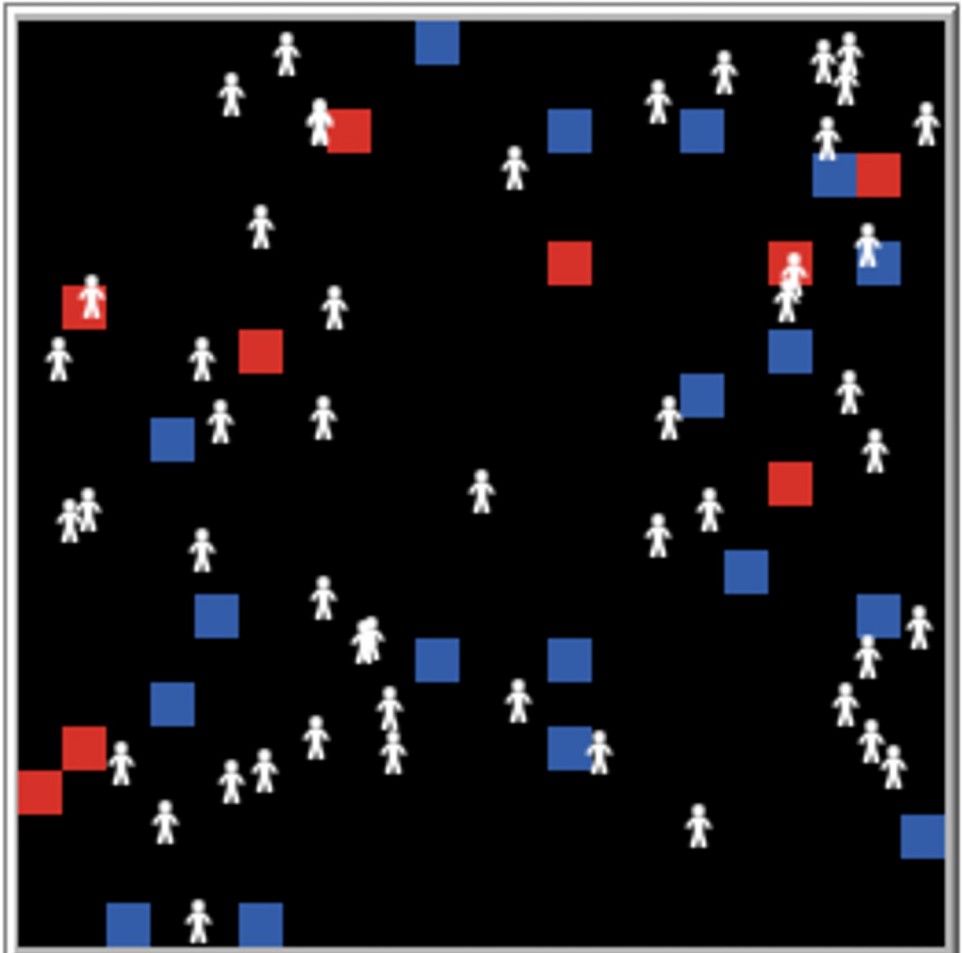

**Fig 7. The developed incentive mechanism simulator.**

The algorithm of the incentive mechanism simulator is shown in Fig 8. The inputs of the algorithm are (1) agent set *AG*, (2) the simulated incentive mechanism instance *NORM*, (3) new agent join rate *joinRate*, and (4) the max simulation ticks *maxTicks*. *AG* and *NORM* are from the generated simulation data. The parameters *joinRate* and *maxTicks* are determined by

```
Algorithm 2 : Mechanism Simulator
Input: AG, joinRate, NORM, maxTicks
Output: AG, POST
 1: tick ← 0
 2: POST ← ∅
 3: while tick < maxTicks do
 4:     AG ← AG ∪ newAgent(joinRate)
 5:     for ag ∈ AG do
 6:         POST ← contribute(ag, POST, NORM)
 7:     end for
 8:     tick ← tick + 1
 9: end while
10: return AG, POST
```

**Fig 8. Mechanism simulator.**

**Algorithm 3 : Agent Contribute Procedure**

**Input:** $ag, POST, NORM$
**Output:** $POST$
1: **if** $pq \geq ag.qr$ **then**
2:     $q \leftarrow$ crateQuestion$(ag)$
3:     $POST \leftarrow POST \cup q$
4: **else**
5:     $q \leftarrow$ select a question from $POST$
6:     **if** pan$(q.n)$*pat$(q.t) > 1 - ag.qr$ **then**
7:         $a \leftarrow$ createAnswer$(q)$
8:         $POST \leftarrow Post \cup a$
9:     **end if**
10: **end if**
11: **if** $pu \geq ag.ur$ **then**
12:     $p \leftarrow$ select a post from $POST$
13:     upvote$(p)$
14:     reward$(ag,$owner$(p),NORM)$
15: **else**
16:     $p \leftarrow$ select a post from $POST$
17:     downvote$(p)$
18:     penalty$(ag,$owner$(p),NORM)$
19: **end if**
20: **return** $POST$

**Fig 9. Agent contribute procedure.**

the users' growth rate of the real community and the days of the simulated community, respectively. Lines 1-2 initialize agents' environment. Line 4 represents new agents continuously joining the simulator. Lines 5-7 represent the agents' contribution process as described in detail in Fig 12. The simulation times are limited by Lines 3 and 8.

The inputs for Fig 9 are contributor $ag$, its environment $POST$, and the incentive mechanism $NORM$. Lines 1-3 describe the questioning procedure of $ag$. Lines 4-11 describe the answering procedure of $ag$. Here, the functions $pan(n)$ and $pat(t)$ denote an agent's probability of answering a question having $n$ answers and a question created $t$ days ago, respectively. Lines 12-17 and Lines 18-24 describe $ag$'s upvoting and downvoting process, respectively.

The source code, the full simulation results in summary and the case studies used for its evaluation are available at the GitHub site (https://www.github.com/snryou/NorMASS).

### 4.6. Capturing simulation emergence

In this step, we run the developed simulator to capture the emergences on the generated simulation data, which are related to community macro attributes or behavior characteristics, as shown in Table 3.

## 5. Experimental evaluation

To evaluate our proposed approach, we use the approach to simulate the Stack Overflow incentive mechanism.

### 5.1. Research questions

To evaluate the performance of the proposed approach, we intend to answer the following questions.

**Table 3. Simulation emergence.**

| Emergence | Description |
|---|---|
| reputation | Distribution of users with different reputations. |
| daily-posts | Daily posts of users with different reputations. |
| daily-votes | Daily votes of users with different reputations. |
| question-rate | Question-rate of users with different reputations. |
| upvote-rate | Upvote-rate of users with different reputations. |
| daily-questions | Daily questions of users with different reputations. |
| daily-answers | Daily answers of users with different reputations. |
| daily-upvotes | Daily upvotes of users with different reputations. |
| daily-downvotes | Daily downvotes of users with different reputations. |
| fast-answers | Distribution of answers of different question ages. |
| question-distribution | Distribution of questions of different answer numbers. |

**RQ1: How well does the data we simulate reflect the real data?** One should be able to run the incentive mechanism model on generated data that reflect the characteristics of the real community. RQ1 aims to determine if the simulation data reflect the fundamentals of the real community.

**RQ2: How well our proposed model reflects the incentives of the community?** A fundamental but important requirement of our simulations is that the simulated effect should be in alignment with those observed through the real community. The goal of RQ2 is to provide confidence that our simulation can reproduce the macro emergence of the real community, including users' attributes, the quantity of contribution, and the speed of contribution.

**RQ3: How consistent can we keep the simulation performance across different language communities?** Our simulation is performed across different language communities. While different communities will inevitably perform differently in simulation due to differences in data, we would expect great consistency in simulation performance using data from different communities. If the performance across different communities is mostly consistent, we can believe that the simulation results are meaningful. The goal of RQ3 is to measure the level of consistency in the simulation performances of incentive mechanisms across different language communities.

## 5.2. Evaluation methodology

**5.2.1. Case studies.** We consider the reputation mechanism of Stack Overflow between Jan. 2017 and Sep. 2019 as our case study, as shown in Table 1. In addition, we downloaded the data of the top five language communities of SO from the brentozar website (http://www.brentozar.com/archive/2015/10/how-to-download-the-stack-overflow-database-via-bittorrent/). The number of posts and users in the language communities on Jan. 1st 2017 is shown in Table 4.

**Table 4. Top 5 language communities of SO on 1 January 2017.**

| Language | Post | User |
|---|---|---|
| JavaScript | 185,232 | 65,441 |
| Python | 158,593 | 45,401 |
| Java | 127,990 | 47,084 |
| C# | 96,268 | 35,452 |
| PHP | 88,438 | 31,910 |

**5.2.2. Experimental setup.**   In response to the RQs, we ran the developed simulator over simulation data. The reputation rules of the simulator are the same as those of SO. The number of agents is one percent of the actual community users on Jan. 1st, 2017. Agent participation rate *joinRate* is the daily user growth rate. The simulation ticks *max-ticks* denote the days from Jan. 1st, 2017 to Sep. 1st, 2019, i.e. 973.

After the simulator runs, new agents continuously join the system and diverse agents contribute according to their behavior rules. Finally, we capture the emergence of the system. To smooth out stochastic errors, we replicate the simulation of each community 5 times using the simulation data. The experiment was performed on a computer with a 2.4GHz dual-core processor and 32GB of memory.

**5.2.3. Evaluation metrics.**   Being a good fit for the actual community, the proposed model demonstrates its strong prediction potential. To evaluate the proposed model, we compare the emergence of the simulator with the counterpart of SO under the same reputation rules. The comparison is measured by the correlation and approximation between them. Let $S$ and $M$ be the emergence of SO and of the simulator, $S_i$ and $M_i$ be their $i$th components, respectively. We evaluate the performance of the proposed model based on Eqs 13–15.

Eq 13 may be solved for Pearson Correlation Coefficient (*pcc*) between two datasets. In other words, the similarity of their data trends can be determined [43]. Eq 14 describes the average value approximation (*ava*) between them. The greater *ava* of two datasets, the closer their values. Eq 15 integrates the effect of the two metrics. The greater the index *sim*, the better the fit of the proposed model.

$$pcc(S, M) = \frac{\sum_{i=1}^{n}(S_i - \bar{S})(M_i - \bar{M})}{\sqrt{(S_i - \bar{S})^2}\sqrt{(M_i - \bar{M})^2}}. \tag{13}$$

$$ava(S, M) = 1 - \frac{1}{n}\sum_{i=1}^{n}\frac{|S_i - M_i|}{\max(S)}. \tag{14}$$

$$sim(S, M) = \frac{pcc(S, M) + ava(S, M)}{2}. \tag{15}$$

## 5.3. Result

**RQ1.** To answer RQ1, we first collect user attribute data from the five language communities of Stack Overflow on reputation, daily post activity, vote activity, question rate, and upvote rate distribution. Then, we run the data generator five times for producing simulation data for per community and average them for analysis. The number of agents is one percent of the users of per actual community on Jan. 1st, 2017. Finally, we use the metric of Eq 8 to compare their difference

Fig 10 presents similarities between the individuals' attributes simulation. As indicated, their similarities are all greater than 0.85 for agents generated by the simulation data algorithm. Among them, the *pac* attribute has the highest simulation similarity, all of which are above 90%. The user questioning preference of PHP community (*qr*) has the smallest, about 85%. The result shows a close agreement between the simulation data and the real users across the five attributes considered.

---

**Answer to RQ1:** The analysis demonstrates that the data produced by the simulation data generator accurately reflects the fundamentals of the real community.

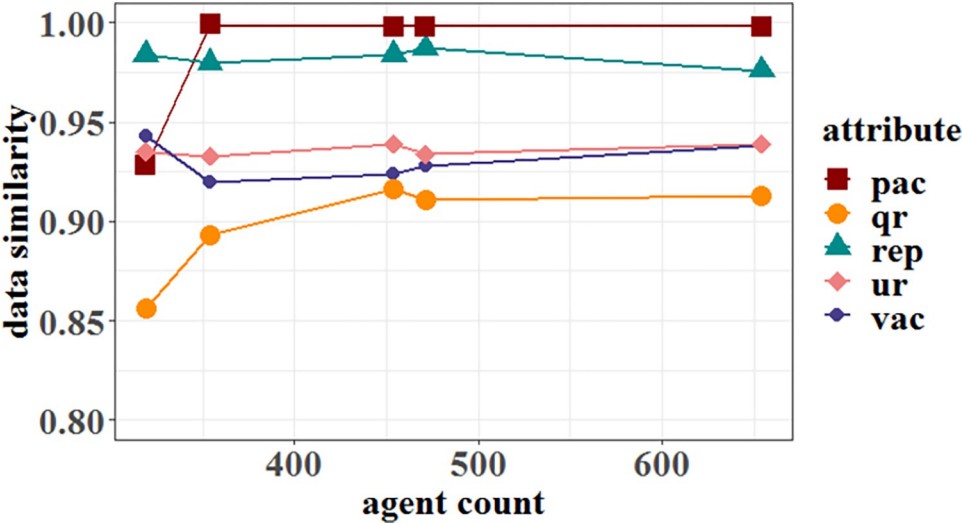

**Fig 10. Experimental data consistency analysis.** The abscissa represents the number of agents generated, which is one percent of the number of users of per actual community. The ordinate represents the similarities between the individuals' attributes simulation in different communities.

**RQ2.** To answer RQ2, we reproduce the emergence of the Java language community between Jan. 1, 2017, and Sep. 1, 2019, including user attribute, answer pattern, and activeness in contribution. As shown in Fig 11, community users are distributed according to the power law. Most of them have a reputation point of less than 10, and very few have a reputation point greater than 120. Our model fits SO user distribution very well (*sim* =0.971).

The left part of Fig 12 shows the simulation of SO users' daily posts and votes. Both demonstrate individuals can be significantly stimulated by their reputation mechanisms. Their simulation similarities reach 0.901 and 0.869, respectively. Our simulation also adequately reproduces the question rate and upvote rate of SO users with different reputation points, as shown on the right of Fig 12. Individuals with higher reputation points exhibit less answering and upvoting behavior than individuals with low reputation points in the two systems. The similarities between them are all 0.937, respectively.

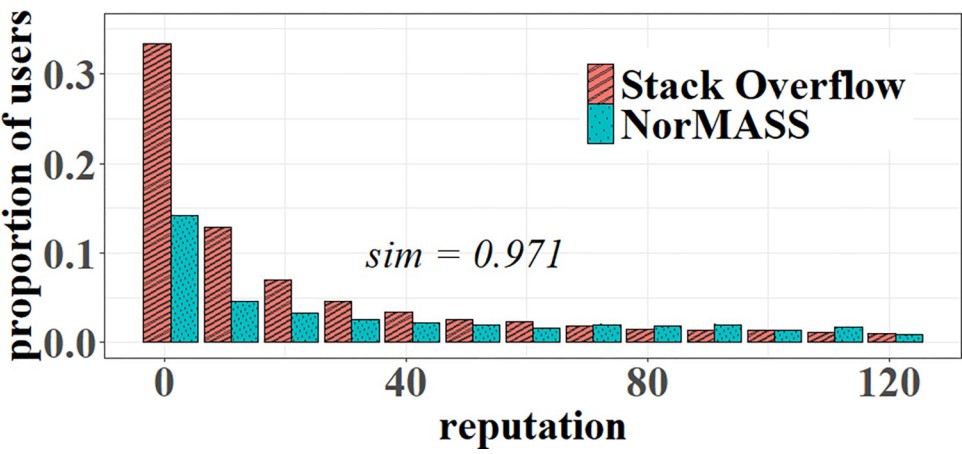

**Fig 11. User reputation distribution simulation.**

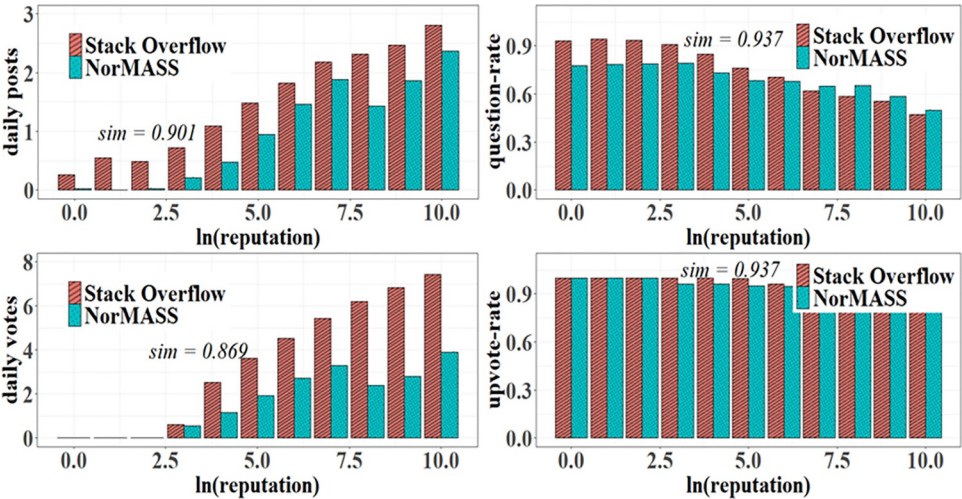

**Fig 12. User attribute emergence simulation.**

In detail, our simulation effectively reproduced the emergence of SO users' contribution behavior, including questioning pattern, answering pattern, upvoting pattern, and downvoting pattern. As shown in Fig 13, individual contribution behavior increases with individuals' reputation points in the two systems. Moreover, individuals of lower reputations seldom answer questions in the two systems. By contrast, individuals of higher reputations create more answers in the two systems on average. Driven by the reputation mechanism, individuals are more likely to vote up posts than to vote down them. The similarities of the simulation of users' contribution behavior are all greater than 0.85.

In Fig 14, we can see that individuals' answering behavior is greatly influenced by question age and the number of answers to questions faced questions for earning more points. The more recent a question is asked and the fewer answers it has, the more individuals are willing to answer the question. The similarities between the two systems are 0.941 and 0.895, respectively.

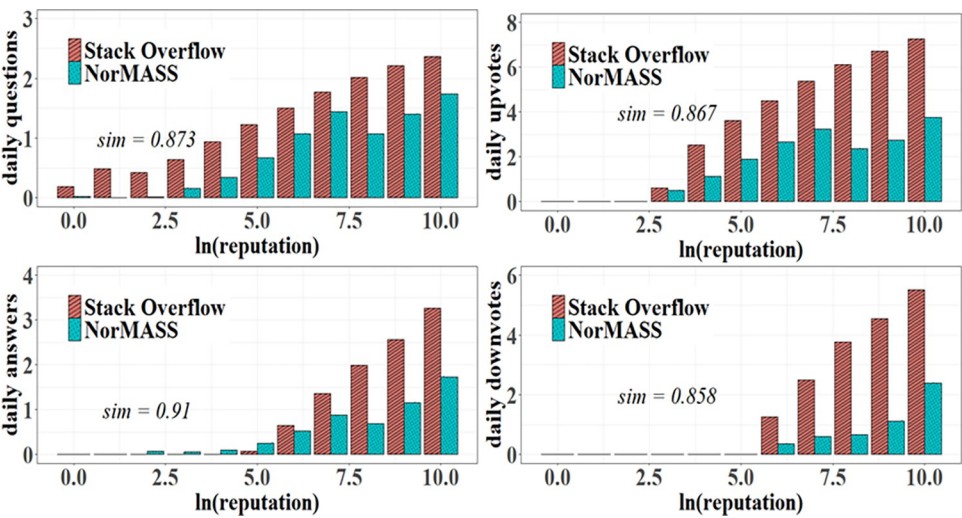

**Fig 13. Contribution activeness simulation.**

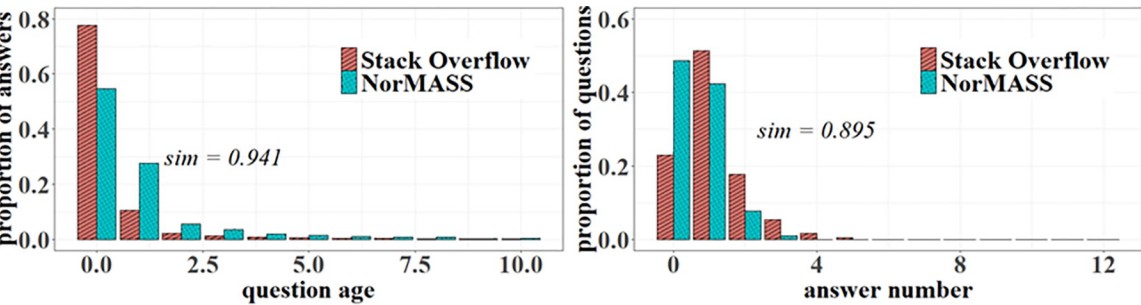

**Fig 14. Response speed and answer distribution simulation.**

**Answer to RQ2:** As shown in Figs 11–14 we reproduced eleven emergences on Java language community of Stack Overflow under the incentive mechanism. Their similarities of simulation are all greater than 0.8. These results suggest that our reproduced emergence is highly approximate to the emergence of the actual community. Thus our proposed model and developed simulator are adequately effective. The proposed model has the potential to simulate the effects of an incentive mechanism on its community.

**RQ3.** To check the consistency of simulation effects across different language communities, we generated simulation data in accordance with the number of users of five language communities on Jan 1, 2017 and ran the simulator to reproduce their emergence. Table 5 shows the results, with *Mean*, *Max*, *Min*, and *Std. Dev* denoting the mean, maximum, minimum, and standard deviation of similarities for the different communities. All similarities of simulations for different language communities are greater than 0.8 and all the deviations are less than 0.06. The summary thus provides us confidence in the consistency of the simulation results across different communities.

It should be noted that there are alternative or complementary approaches for testing the similarity such as information entropy-based metrics [44] and the mean relative error criterion [45]. In view of the definite results from the similarity test (sim > 0.8 everywhere), we did not perform further tests.

**Answer to RQ3:** The result in Table 5 gives evidence that the proposed approach can keep consistent simulation performance across different language communities.

**Table 5. Simulation result on the five communities.**

| Emergence | Mean | Max | Min | St. Dev |
|---|---|---|---|---|
| reputation | 0.919 | 0.971 | 0.815 | 0.058 |
| daily-posts | 0.892 | 0.904 | 0.879 | 0.011 |
| daily-votes | 0.866 | 0.871 | 0.854 | 0.007 |
| question-rate | 0.892 | 0.937 | 0.832 | 0.043 |
| upvote-rate | 0.946 | 0.956 | 0.937 | 0.009 |
| daily-questions | 0.867 | 0.878 | 0.852 | 0.013 |
| daily-answers | 0.911 | 0.916 | 0.905 | 0.004 |
| daily-upvotes | 0.864 | 0.869 | 0.851 | 0.007 |
| daily-downvotes | 0.867 | 0.882 | 0.858 | 0.010 |
| fast-answers | 0.939 | 0.943 | 0.935 | 0.003 |
| question-distribution | 0.889 | 0.902 | 0.866 | 0.016 |

## 6. Conclusion

In this work, we proposed a normative MAS-based approach for simulating incentive mechanisms. The approach (NorMASS) includes a normative MAS-based formal simulation model, the formal representation of agent behaviors, an automated simulation data generator, and an open-source tool to provide the basic prototype model for reusing. Using a reputation mechanism from Stack Overflow, we performed an empirical evaluation demonstrating that the proposed approach has the capability of simulating the effect of incentive mechanisms and that these mechanisms are in alignment with the actual community. Our approach provides indications for community managers to predict the effect of incentive mechanisms, and inspirations for researchers to explore the emergence of complex socio-technical systems such as Q&A communities.

However, there are three strands of limitations in the proposed approach. First, we did not consider the role of incentive mechanisms in regulating the quality of users' contributions. The equations in Section 4.3 were designed for describing the relationship between the agents' attributes. These equations cannot account for the role of incentive mechanisms in regulating the quality of users' contributions; anyway, incentive mechanism exerts little effect on the quality of users' contribution [46]. Therefore, the equations need to be further improved if it is applied to explore the impact of incentive mechanisms on community post quality.

Second, we did not consider posts in simulation data generation. The initial generated simulation data contains only users' information, without considering the simulation of their historical posts. In the real community, there exists a certain possibility for users, though small in number answer or vote for these long-ago posts. All of these bring forth some of our simulation errors.

Third, we have no evidence to show users' contribution behavior may only be influenced by the reputation mechanism at a given time. A thorough investigation is required to determine the extent to which our approach is useful for multiple incentives in the actual Q&A communities, e.g., badges and privileges.

In future studies, we will improve our approach by considering the quality of users' contributions, the generation of post simulation data, and the roles of multiple incentive mechanisms of Q&A communities. Moreover, we will extend and evaluate incentive prediction systems in other virtual communities. More importantly, we will use our approach to guide the design of the Q&A community incentive mechanism.

## Author Contributions

**Conceptualization:** Yi Yang, Xinjun Mao.

**Data curation:** Yi Yang.

**Formal analysis:** Yi Yang, Xinjun Mao.

**Funding acquisition:** Xinjun Mao.

**Methodology:** Yi Yang.

**Software:** Yi Yang.

**Validation:** Yi Yang.

**Visualization:** Yi Yang.

**Writing – original draft:** Yi Yang.

**Writing – review & editing:** Yi Yang, Xinjun Mao, Shuo Yang, Menghan Wu.

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
