## [Decision Letter · Decision Letter 0]

4 Oct 2022

PONE-D-22-24187NorMASS: A normative MAS-based modeling approach for simulating incentive mechanisms of Q&A communitiesPLOS ONE

Dear Dr. Yang,

Thank you for submitting your manuscript to PLOS ONE. After careful consideration, we feel that it has merit but does not fully meet PLOS ONE’s publication criteria as it currently stands. Therefore, we invite you to submit a revised version of the manuscript that addresses the points raised during the review process.

We look forward to receiving your revised manuscript.

Kind regards,

Le Zhang

Academic Editor

PLOS ONE

Journal Requirements:

2. In your Methods section, please include additional information about your dataset and ensure that you have included a statement specifying whether the collection and analysis method complied with the terms and conditions for the source of the data.

Please note that PLOS ONE has specific guidelines on code sharing for submissions in which author-generated code underpins the findings in the manuscript. In these cases, all author-generated code must be made available without restrictions upon publication of the work. Please review our guidelines at https://journals.plos.org/plosone/s/materials-and-software-sharing#loc-sharing-code and ensure that your code is shared in a way that follows best practice and facilitates reproducibility and reuse.

Reviewers' comments:

Reviewer's Responses to Questions

**Comments to the Author**

1. Is the manuscript technically sound, and do the data support the conclusions?

Reviewer #1: Yes

Reviewer #2: Yes

2. Has the statistical analysis been performed appropriately and rigorously? 

Reviewer #1: Yes

Reviewer #2: Yes

3. Have the authors made all data underlying the findings in their manuscript fully available?

Reviewer #1: Yes

Reviewer #2: Yes

4. Is the manuscript presented in an intelligible fashion and written in standard English?

Reviewer #1: Yes

Reviewer #2: Yes

5. Review Comments to the Author

Reviewer #1: I carefully read the manuscript “NorMASS: A normative MAS-based modeling approach for simulating incentive mechanisms of Q&A communities”, which proposed a Normative Multi-Agent System based Simulation approach to simulate community emergence.

This is a well written paper containing some interesting results. But there still are some shortcomings that need to be addressed：

1. This is a well-organized paper. However, I do not think it is necessary to divide it into 7 sections. For example, section 7 “Conclusion” could be integrated with section 5, and I suggest that the content of section 6 "Related Work" could be described in the section “Introduction”.

2. The figures in the manuscript are quite small and not very clear, especially Figure 8 - Figure 11.

3. The equations in the manuscript are numbered from 3.3.1，however，I suggest that all the equations should be numbered, which means starting from subsection 3.2.3.

Reviewer #2: The paper proposed a normative MAS-based approach for simulating incentive mechanisms. The work is very interesting, but I have a few small questions：

1、Could the authors consider comparing real data from multiple communities instead of simulated data?

2、Could you add more explanation to section 4.3

3、There are some obvious spelling mistakes in the article.

6. PLOS authors have the option to publish the peer review history of their article (what does this mean?). If published, this will include your full peer review and any attached files.

Reviewer #1: No

Reviewer #2: No

---

## [Author Response · Author response to Decision Letter 0]

7 Nov 2022

Dear Reviewers,

Thank you for giving us the opportunity to submit a revised draft of the manuscript “NorMASS: A normative MAS-based modeling approach for simulating incentive mechanisms of Q&A communities” for publication in the Journal of PLOS ONE. We appreciate the time and effort that you and the reviewers dedicated to providing feedback on our manuscript and are grateful for the insightful comments on and valuable improvements to our paper. We have incorporated most of the suggestions made by the reviewers. Those changes are highlighted in the file labeled “Revised Manuscript with Track Changes”. 

Best regards,

Yi Yang and Xinjun Mao.

---

## [Editor Report · Decision Letter 1]

24 Jan 2023

NorMASS: A normative MAS-based modeling approach for simulating incentive mechanisms of Q&A communities

PONE-D-22-24187R1

Dear Dr. Yang,

We’re pleased to inform you that your manuscript has been judged scientifically suitable for publication and will be formally accepted for publication once it meets all outstanding technical requirements.

Kind regards,

Le Zhang

Academic Editor

PLOS ONE
---

## [Editor Report · Acceptance letter]

27 Jan 2023

PONE-D-22-24187R1 

NorMASS: A normative MAS-based modeling approach for simulating incentive mechanisms of Q&A communities 

Dear Dr. Yang:

I'm pleased to inform you that your manuscript has been deemed suitable for publication in PLOS ONE. Congratulations! Your manuscript is now with our production department. 

Kind regards, 

on behalf of

Dr. Le Zhang 

Academic Editor

PLOS ONE